# Maintenance of age in human neurons generated by microRNA-based neuronal conversion of fibroblasts

Christine J Huh[1,2], Bo Zhang[1], Matheus B Victor[1,3], Sonika Dahiya[4], Luis FZ Batista[1,5], Steve Horvath[6,7], Andrew S Yoo[1]*

[1]Department of Developmental Biology, Washington University School of Medicine, St. Louis, United States; [2]Program in Molecular and Cellular Biology, Washington University School of Medicine, St. Louis, United States; [3]Program in Neuroscience, Washington University School of Medicine, St. Louis, United States; [4]Department of Pathology and Immunology, Washington University School of Medicine, St. Louis, United States; [5]Department of Medicine, Washington University School of Medicine, St. Louis, United States; [6]Department of Human Genetics, David Geffen School of Medicine, University of California, Los Angeles, Los Angeles, United States; [7]Department of Biostatistics, Fielding School of Public Health, University of California, Los Angeles, Los Angeles, United States

*For correspondence: yooa@wustl.edu

Competing interests: The authors declare that no competing interests exist.

**Abstract** Aging is a major risk factor in many forms of late-onset neurodegenerative disorders. The ability to recapitulate age-related characteristics of human neurons in culture will offer unprecedented opportunities to study the biological processes underlying neuronal aging. Here, we show that using a recently demonstrated microRNA-based cellular reprogramming approach, human fibroblasts from postnatal to near centenarian donors can be efficiently converted into neurons that maintain multiple age-associated signatures. Application of an epigenetic biomarker of aging (referred to as epigenetic clock) to DNA methylation data revealed that the epigenetic ages of fibroblasts were highly correlated with corresponding age estimates of reprogrammed neurons. Transcriptome and microRNA profiles reveal genes differentially expressed between young and old neurons. Further analyses of oxidative stress, DNA damage and telomere length exhibit the retention of age-associated cellular properties in converted neurons from corresponding fibroblasts. Our results collectively demonstrate the maintenance of age after neuronal conversion.

## Introduction

Increasing evidence suggests that in addition to genetic susceptibility, age-related neurodegeneration may be caused in part by cellular aging processes that result in accumulation of damaged DNA and proteins in neurons (*Mattson and Magnus, 2006*). However, because of the inaccessibility to neurons from elderly individuals, studying these age-related cellular processes in human neurons remains a difficult task. Cellular reprogramming approaches have explored generating populations of human neurons by inducing pluripotent stem cells (iPSC) from human fibroblasts and subsequent differentiation into neurons (*Takahashi and Yamanaka, 2006*; *Takahashi et al., 2007*; *Hanna et al., 2008*; *Hu et al., 2010*). Importantly, this induction of pluripotency in adult fibroblasts reverts cellular age to an embryonic stage (*Lapasset et al., 2011*; *Patterson et al., 2012*) which remains even after differentiation into neurons (*Patterson et al., 2012*; *Miller et al., 2013*). While this is useful for

**eLife digest** As we age, so do our cells. When cells are used in the laboratory to study the biology of diseases, it is important that the age of the cells reflects the age at which the disease develops. This is particularly important for illnesses with symptoms that develop during old age, and where younger cells may appear to be relatively unaffected.

Aging is a major risk factor in many brain disorders that affect elderly individuals. These late-onset disorders can be difficult to study because it is rarely possible to collect diseased cells from patients. Recent experimental advances, however, now mean that unrelated cell types – typically cells called fibroblasts, taken from a patient's skin – can be converted directly into brain cells instead. These new brain cells will have the same genetic makeup as the patients they came from, but whether these converted cells would reflect the patient's age too remained to be determined.

By measuring a range of biological properties of the converted cells, Huh et al. now show that converted cells do indeed keep track of their age when they are changed from fibroblasts to brain cells. The age of the cells was tested by looking at age-linked markers attached to their DNA known as an "epigenetic clock". In addition, Huh et al. measured the age of the cells by examining the expression of genes altered with aging. Other factors examined included the amount of damaged DNA and the size of DNA regions called telomeres, which become shorter with age. Together, all of these indicated that the converted brain cells retain the age of the fibroblasts that they were made from.

So far this work has only been done using fibroblasts collected from healthy people. The same tests now need to be done using cells from people with late-onset illnesses like Huntington's disease and Alzheimer's disease. If the converted brain cells show signs of illness, it may provide new ways to study these illnesses using cells from specific patients, which may eventually lead to the development of new treatments.

modeling early developmental phenotypes (*Lafaille et al., 2012*; *Lee et al., 2009*), iPSC-derived cells have been reported to be unsuitable in recapitulating phenotypes selectively observed in aged cells (*Mattson and Magnus, 2006*; *Vera and Studer, 2015*). Recently, experimental manipulations to accelerate aging in iPSC-derived cells have been explored, for instance, by overexpressing progerin, a mutant form of lamin A observed in progeria syndrome, to force the detection of age-related pathophysiology of neurodegenerative disease (*Arbab et al., 2014*; *Cornacchia and Studer, 2015*).

Alternatively, we previously described a reprogramming paradigm using neuronal microRNAs (miRNAs), miR-9/9* and miR-124 (miR-9/9*-124), that exert reprogramming activities to directly convert human fibroblasts to specific mature neuronal subtypes (*Richner et al., 2015*; *Victor et al., 2014*; *Yoo et al., 2011*). Because this neuronal conversion is direct and bypasses pluripotent/multipotent stem cell stages, we reasoned that miR-mediated directly reprogrammed neurons would retain the age signature of the original donor. To assess the cellular age, we used the epigenetic clock method, which is a highly accurate biomarker of age based on DNA methylation (*Horvath, 2013*). Further, we evaluated age-associated signatures based on gene expression levels, miRNAs, and cellular readouts considered to be hallmarks of aging (*López-Otín et al., 2013*). Our thorough investigation into multiple age-associated signatures collectively demonstrate the maintenance of cellular age of the original donor during neuronal conversion and strongly suggest that directly converted human neurons can be advantageous for studying age-related neuronal disorders.

## Results and discussion

### MiRNA-mediated neuronal conversion across the age spectrum

Establishing robust reprogramming efficiency is essential prior to assessing age-related phenotypes in reprogrammed neurons. We therefore elected to test our recently developed conversion approach that utilizes miR-9/9*-124 and transcription factors to robustly generate a highly enriched

population of striatal medium spiny neurons (MSNs) from the fibroblasts of donors of varying ages (*Richner et al., 2015*; *Victor et al., 2014*). Fibroblast samples from donors ranging from 3 days to 96 years of age were collected and expanded to match population doubling level (PDL) to eliminate any confounding variability introduced by sequential passaging (*Campisi and d'Adda di Fagagna, 2007*; *Pazolli and Stewart, 2008*), then subsequently transduced with miR-9/9*-124 with CTIP2, DLX1/2, and MYT1L (CDM) following our established protocol (*Richner et al., 2015*; *Victor et al., 2014*) (*Figure 1A*). Reprogrammed cells were then stained for neuronal markers, MAP2, TUBB3, NeuN and MSN markers, DARPP32 and GABA (*Figure 1B–C*). MAP2 and TUBB3-positive reprogrammed neurons exhibiting extensive neurite outgrowth represented approximately 70–80% of the cell population (*Figure 1D*), demonstrating the consistency of reprogramming efficiency in all fibroblast samples. Neurons reprogrammed from young and old fibroblasts exhibited fast inward currents and action potentials in monocultures without necessitating coculturing with glial cells or primary neurons (n = 11) (*Figure 1E*). In addition, the consistent upregulation of neuronal genes, including MAP2, NCAM, and voltage-gated sodium channels, and downregulation of fibroblast-associated genes were observed in reprogrammed neurons from both young and old cells (*Figure 1—figure supplement 1*). These results suggest the consistent applicability of miRNA-based neuronal reprogramming in fibroblasts of all ages.

## Maintenance of epigenetic age during neuronal conversion

Aging largely influences the epigenetic landscape of cells (*Oberdoerffer and Sinclair, 2007*) with a number of genomic loci becoming differentially methylated with age (*Horvath et al., 2012*; *Christensen et al., 2009*). The epigenetic clock, which analyzes the methylation status of 353 specific CpG loci, has been shown to be a highly accurate age estimator that applies to all human organs, tissues, and cell types (*Horvath, 2013*). The epigenetic clock leads to an age estimate (in units of years) which is referred to as epigenetic age or DNA methylation (DNAm) age. We analyzed DNA methylation levels of neurons converted from 16 fibroblast samples aged three days to 96 years alongside 37 fibroblast samples aged three days to 94 years (*Figure 2A*). The actual chronological donor age was highly correlated with the estimated DNAm age of fibroblasts (correlation = 0.75) (*Figure 2B*) and of reprogrammed neurons (correlation = 0.82) (*Figure 2B*). Importantly, when the DNAm age of each reprogrammed neuron was compared to the DNAm age of the corresponding fibroblast, there was a near-perfect correlation (correlation = 0.91), which suggests that the epigenetic clock is unperturbed during miRNA-based neuronal reprogramming (*Figure 2C*) and supports the notion of age maintenance during direct neuronal conversion. By contrast, it is known that iPSC generation resets the epigenetic clock to an embryonic state since iPSCs have a DNAm age that is negative or close to zero (*Horvath, 2013*).

## 'Aging' transcriptome and microRNAs in reprogrammed neurons

Given the broad variability in gene expression with age in multiple cell types (*Berchtold et al., 2008*; *Lu et al., 2004*; *Fraser et al., 2005*; *Glass et al., 2013*) and a recent demonstration of maintenance of age-associated transcriptomic changes in neurons directly converted by transcription factors (*Mertens et al., 2015*), we sought to determine whether age-associated transcriptomic changes could be detected after miR-9/9*-124-CDM-based neuronal conversion. The transcriptome and microRNA profiles of reprogrammed neurons from both young and old fibroblasts were analyzed alongside corresponding fibroblasts. Principal component analysis (PCA) of transcriptome revealed cell type-specific clustering of reprogrammed neurons versus fibroblasts, while age-associated segregation is observed in both fibroblasts and reprogrammed neurons (*Figure 3A*). A cohort of upregulated and downregulated genes with aging was commonly observed in both fibroblasts and converted neurons (*Figure 3B*), consistent with a previous report (*Mertens et al., 2015*). Gene ontology (GO) analysis of differentially expressed genes with age in reprogrammed neurons is enriched for terms associated with age-related biological processes (*Figure 3—figure supplement 1*), including vesicle-mediated transport (*Wilmot et al., 2008*), nervous system development (*Lu et al., 2004*) NF-kappaB transcription factor activity (*Tilstra et al., 2011*), regulation of apoptosis and inflammatory response (*de Magalhães et al., 2009*), and for genes previously identified to be associated with age in the human brain (*Lu et al., 2004*).

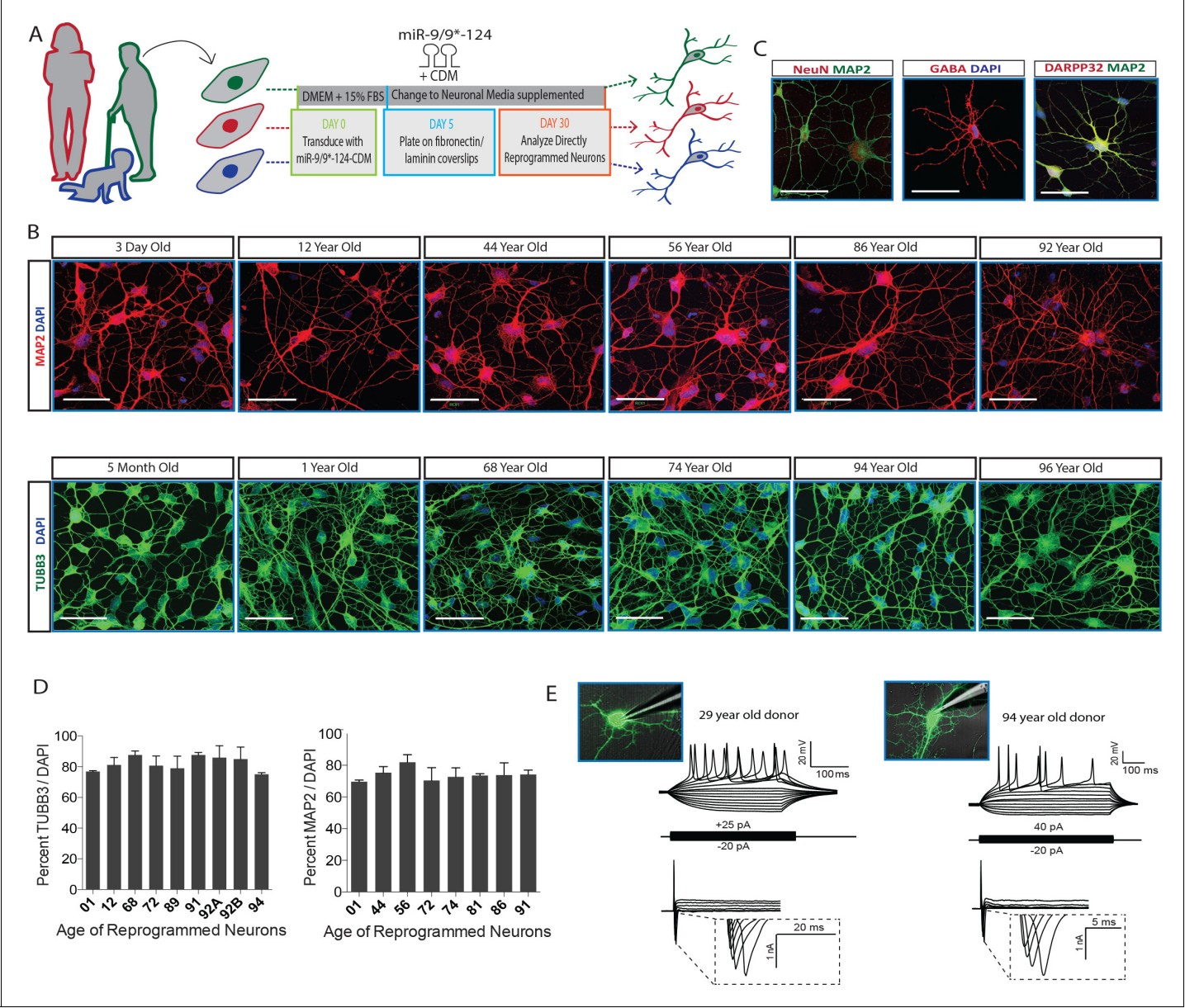

**Figure 1.** MicroRNA-mediated direct neuronal conversion applied to fibroblasts across the age spectrum. (**A**) Schematic diagram of neuronal conversion of human fibroblast samples from individuals ranging from three days to 96 years of age. Primary fibroblasts were transduced with microRNA-9/9*-124 and the transcription factor cocktail CTIP2, DLX1/2, MYT1L (CDM) and analyzed for neuronal properties after 30 days. (**B**) Expression of pan-neuronal markers MAP2 (top) and TUBB3 (bottom) after neuronal conversion of human fibroblasts ranging in age. Scale bar = 50 μm. (**C**) Expression of pan-neuronal marker NeuN and medium spiny neuron-specific markers GABA and DARPP32 in reprogrammed neurons from fibroblasts aged 91-, 72-, 92-years respectively. (**D**) Immunostaining analysis of percentage of reprogrammed neurons positive for neuronal markers TUBB3 and MAP2 over DAPI signals (n = 200–300 per cell line). (**E**) Representative whole-cell current clamp recording of converted neurons from young (29 years old, left) and old (94 year old donor, right) donors. Converted human neurons in monoculture displayed multiple action potentials in response to step current injections at four weeks post-transduction. All reprogrammed neurons from old fibroblasts recorded (n = 11) fired APs in response to current injections (top). Representative traces of fast-inactivating inward currents recorded in voltage-clamp mode. Voltage steps ranged from +10 to +70 mV (bottom).

The following figure supplement is available for figure 1:

**Figure supplement 1.** Transcriptome analyses between converted neurons and fibroblasts of young and old age groups.

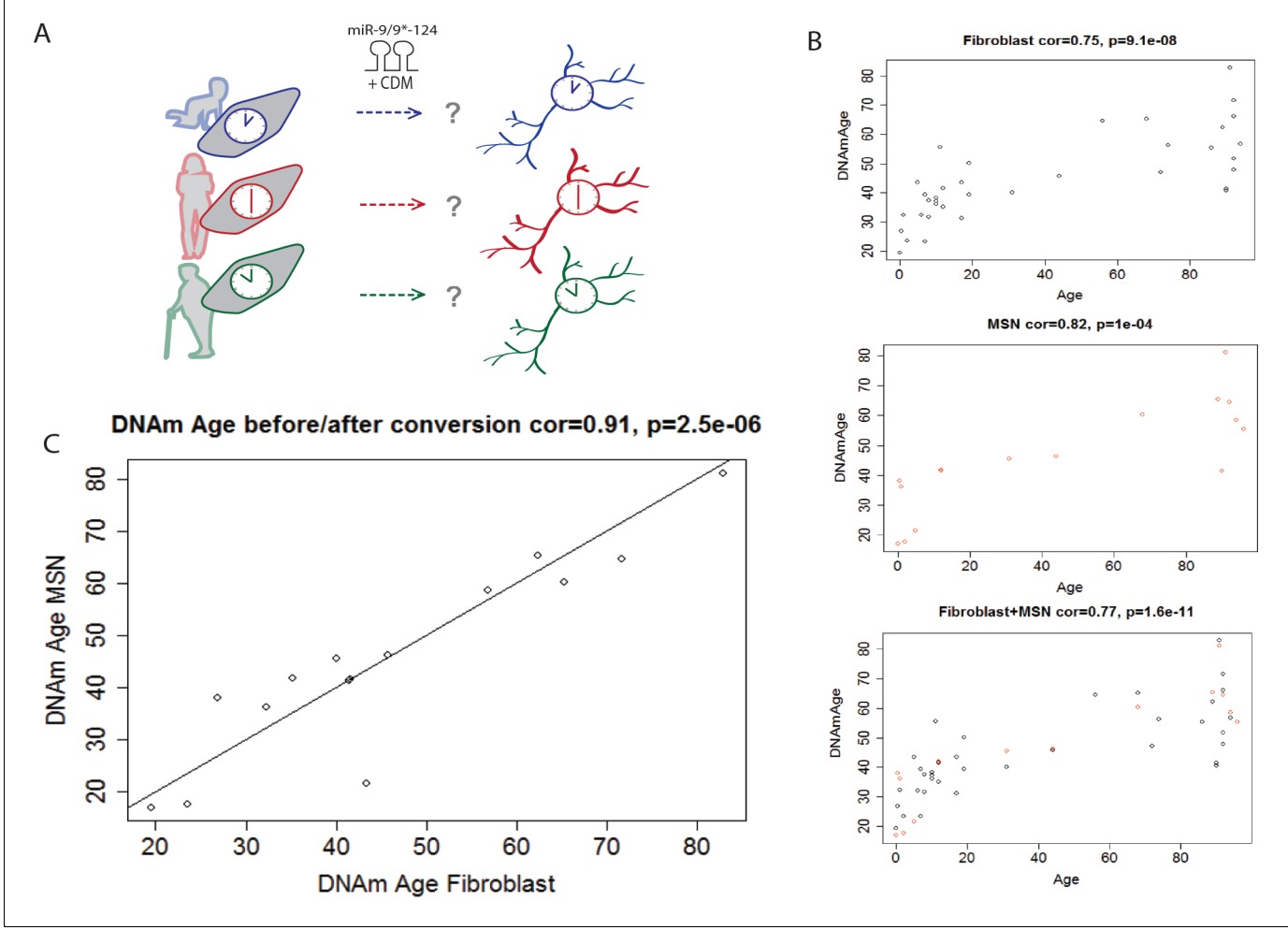

**Figure 2.** Conservation of the epigenetic clock of reprogrammed neurons from human fibroblasts. (**A**) Schematic diagram representing the hypothesis that the epigenetic clock of fibroblasts from different age groups is conserved in reprogrammed neurons after miR-mediated neuronal conversion. (**B**) Top: Predicted ages based on the methylation status (DNAm age) of fibroblasts plotted against the actual ages of fibroblasts (correlation = 0.75, p=9.1e-08). Middle: Predicted DNAm ages of reprogrammed neurons against actual ages of starting fibroblast, correlation = 0.82, p=1e-04. Bottom: Combined plot of DNAm ages of fibroblasts and DNAm ages of reprogrammed neurons against actual ages (correlation = 0.77, p=1.6e-11). (**C**) DNAm age of reprogrammed neurons plotted against the DNAm ages of the corresponding, starting fibroblast ages, correlation = 0.91 p=2.5e-06.

The following source data is available for figure 2:

**Source data 1.** Output for sample information and DNAm ages for fibroblasts and reprogrammed neurons compared to original age.

MicroRNA profiling similarly revealed distinct sample segregation both with age (young versus old) and cell type (fibroblasts versus reprogrammed neurons) (*Figure 3C*). Interestingly, we detected fourteen microRNAs commonly regulated with age in both fibroblasts and reprogrammed neurons, including miR-10a, miR-497, and miR-195, whose expression increased with age (*Figure 3D*). Because microRNAs have been implicated as global regulators of aging-associated cellular processes (*Liu et al., 2012*; *Harries, 2014*) through repression of existing target transcripts (*He and Hannon, 2004*; *Pasquinelli, 2012*; *Lewis et al., 2005*), we reasoned that these age-upregulated microRNAs may target and repress classes of genes found to be downregulated in old reprogrammed neurons. Indeed, GO analysis of predicted targets amongst age-downregulated transcripts in reprogrammed neurons (*Figure 3B*) for miR-10a-5p and miR-497-5p revealed terms associated with age such as metabolism and cellular death and survival (*Figure 3—figure supplement 2*). Our results suggest the potential role of miR-10a and miR-497 in regulating genes involved in cell death

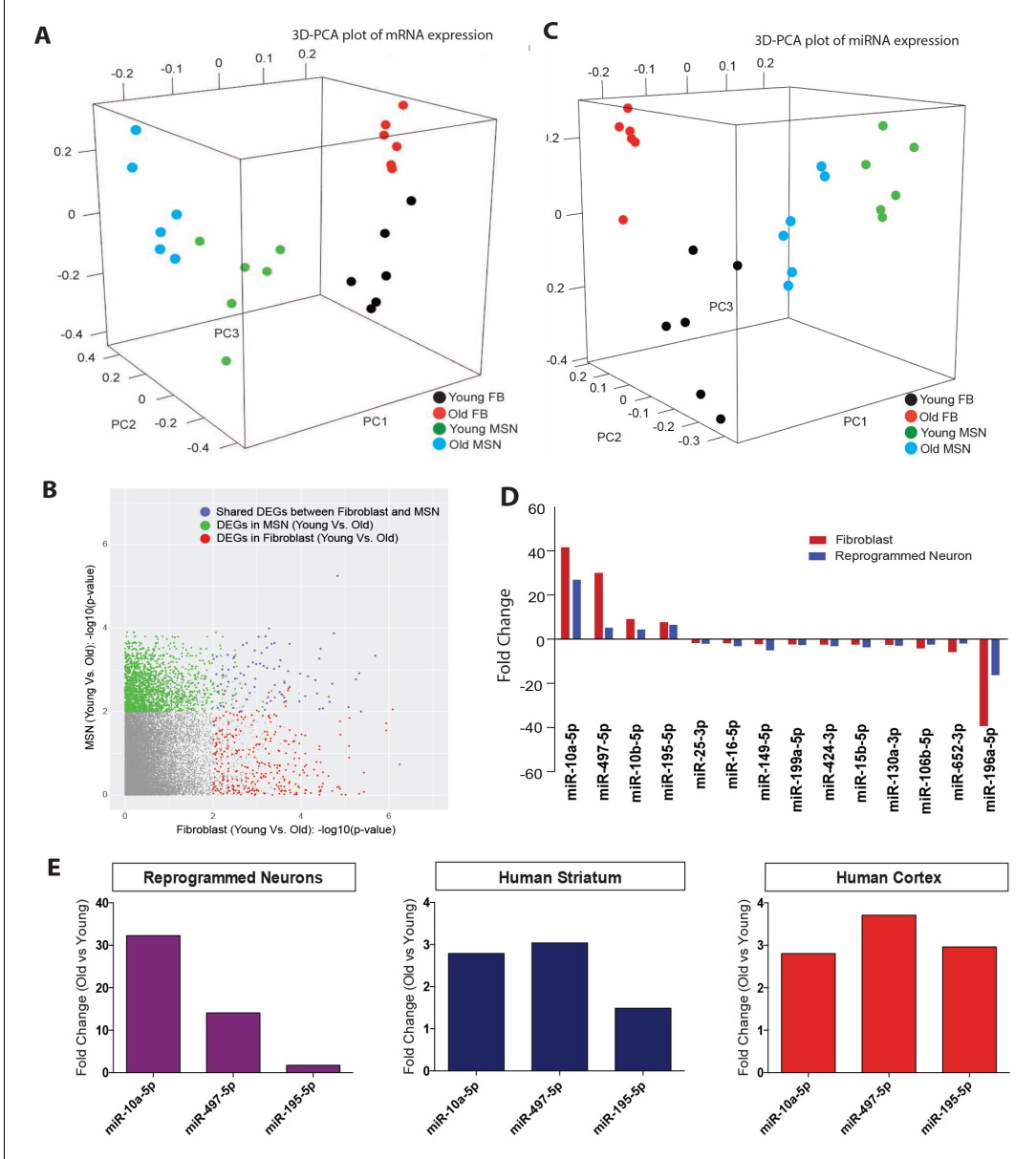

**Figure 3.** Age-associated changes in transcriptome and microRNA profiles in reprogrammed neurons. (**A**) Principle component analysis (PCA) of transcriptome profiling of reprogrammed neurons from young fibroblasts aged three days, five months one year (green) and from old fibroblasts aged 90, 92, and 92 years (blue) alongside corresponding young fibroblasts (black) and old fibroblasts (red) (FDR < 0.05). (**B**) Differentially expressed genes (DEGs) with age in fibroblasts (x axis) and in reprogrammed neurons (y axis). Age-associated DEGs in reprogrammed neurons shown in green, DEGs in fibroblasts shown in red, and commonly shared DEGs with age in both fibroblasts and reprogrammed neurons shown in blue. (**C**) PCA of miRNA profile in reprogrammed neurons from young fibroblasts aged three days, five months one year (green) and from old fibroblasts aged 90, 92, and 92 years (bue) alongside the corresponding young fibroblasts (black) and old fibroblasts (red) (FDR < 0.05). (**D**) MicroRNAs that are differentially regulated with age in both fibroblasts (red) and reprogrammed neurons (blue). Four microRNAs, miR-10a, miR-497, miR-10b, and miR-195, are upregulated with age, while 10 microRNAs are shown to be downregulated with age, p<0.05. (**E**) Validation of expression of miRNA expression upregulated with aging (miR-10a, miR-497, miR-195) in reprogrammed neurons from old fibroblasts over reprogrammed neurons from young fibroblasts (left). Validation of expression changes of miR-10a, miR-497, miR-195 in human striatum slices (middle) and human cortex slices (right) from old individuals compared to young individuals.

The following source data and figure supplements are available for figure 3:

**Source data 1.** Raw data for qPCR for microRNA expression analysis.

**Source data 2.** Full GO terms for age-regulated genes in reprogrammed neurons and for predicted targets of miR-10a-5p and miR-497-5p.

*Figure 3 continued on next page*

*Figure 3 continued*

**Figure supplement 1.** Gene ontology of DEGs in reprogrammed neurons.

**Figure supplement 2.** Gene ontology of predicted targets of age-resulted microRNAs.

and survival, metabolic pathways, and DNA repair. While the exact role of these microRNAs in aging is unknown, miR-10a and miR-497 have been previously implicated in aging-associated cellular processes including inflammation, senescence, metabolism and telomerase activity (*Kondo et al., 2016*; *Qin et al., 2012*; *Fang et al., 2010*). Importantly, the increased expression of miR-10a, miR-497 and miR-195 detected in reprogrammed neurons was also validated by qPCR to be concordantly upregulated in human striatum and cortex samples from old individuals in comparison to young individuals (*Figure 3E*). These results further support the validity of reprogrammed neurons for detecting age-associated changes in microRNA network, mirroring changes observed in human brain.

## Cellular biomarkers reveal maintenance of age in reprogrammed neurons

Directly converted neurons were additionally assayed for cellular hallmarks of aging, including oxidative stress, DNA damage and telomere erosion (*López-Otín et al., 2013*). Oxidative stress has been reported to increase with age, in part due to the accumulation of reactive oxygen species (ROS) (*Keating, 2008*; *Prigione et al., 2010*; *Suhr et al., 2010*; *Cui et al., 2012*). FACS analysis of ROS levels using fluorescent marker MitoSOX (*Miller et al., 2013*) revealed that old reprogrammed neurons have increased ROS levels compared to young reprogrammed neurons, mirroring the observed age-associated differences of ROS levels in fibroblasts (*Figure 4A*). Moreover, reprogrammed neurons were analyzed by Comet Assay, a single-cell gel electrophoresis technique that assesses DNA damage accumulation (*Singh et al., 1990*). Old reprogrammed neurons were found to have longer comet tail lengths, an age-associated property also observed in fibroblasts, that reflects more extensive DNA damage accumulation compared to young cells (*Figure 4B*). Additionally, neuronal conversion maintained the length of telomeres from starting fibroblast which is virtually unchanged (*Figure 4C*), in contrast to the progressive increase in length commonly observed with iPSC reprogramming, where telomeres reach a plateau of around 12–14 kilobases after a few cellular passages (*Agarwal et al., 2010*; *Batista et al., 2011*; *Marion et al., 2009*). Together, these cellular assays support the maintenance of aging marks in old reprogrammed neurons after direct conversion.

## Conclusion

Our in-depth analyses of multiple age signatures provide evidence that neuronal conversion from human somatic cells sampled at different ages generates neurons that emulate the donors' ages. In addition to the demonstration of age-associated transcriptomic changes reported previously (*Mertens et al., 2015*), our results provide novel insights into multiple key signatures associated with age— epigenetic, microRNA and cellular— that are consistently maintained in directly converted neurons. Because aging is a complex process affecting many hallmarks of a cell (*López-Otín et al., 2013*), our assessment of a broad spectrum of age-related markers suggests that directly converted neurons may serve as an alternative model of neuronal aging to iPSC-derived neurons, whose erasure of multiple aging-associated signatures precludes it from adequately modeling, especially, late-onset diseases. Whereas, future studies may investigate whether additional aging marks are similarly conserved in reprogrammed neurons to model different facets of aging. While miR-9/9*-124-based reprogramming can directly convert fibroblasts with similar efficiency to neurons, we note that the older fibroblasts have lower replicative potential in a culture. However, this does not impede in the conversion efficiency of old fibroblasts. MiRNA-mediated generation of aged neurons paves the road to direct conversion into specific neuronal subtypes to investigate the contribution of neuronal aging to late-onset neurodegenerative disorders.

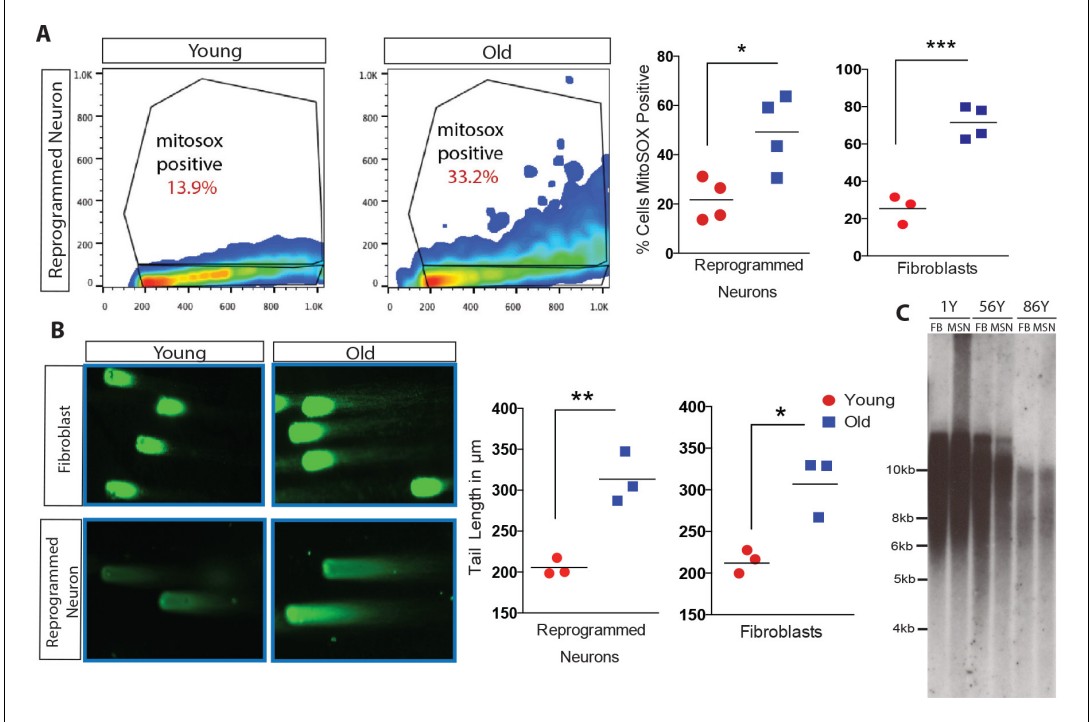

**Figure 4.** Analysis of cellular biomarkers of age reveals conservation of neuron-specific aging in reprogrammed neurons. (**A**) ROS levels visualized by MitoSOX and analyzed by FACs. Representative dot plot of reprogrammed neuron from 1-year-old fibroblast (left) and from 91-year-old fibroblast (right), y axis = FL2 channel (MitoSOX), x axis = FSC. Quantification of percent of cells positive for MitoSOX reveals a significant difference in fibroblasts with age in addition to reprogrammed neurons with age. ***p-value=0.0008; *p-value=0.019. (**B**) Representative images of comets indicating DNA damage from five month old fibroblasts (top left) versus old fibroblasts (top right) alongside reprogrammed neurons from 5-month-old fibroblasts (bottom left) and 92-year-old fibroblasts (bottom right). Quantification of tail lengths in fibroblasts and reprogrammed neurons with age *p-value=0.013, **p-value=0.004. (**C**) Telomere length analyses of reprogrammed neurons are maintained from corresponding starting fibroblasts from one year old, 56 year old, and 86 year old donors.

The following source data is available for figure 4:

**Source data 1.** Raw data for mitoSOX and comet assay.

# Materials and methods

## Cell culture, cell lines and population doubling level (PDL) matching

The following fibroblast cell lines ranging in age from three day old to 96 year old were obtained from the NIA Aging Cell Repository at the Coriell Institute for Medical Research, Coriell ID, RRID#: AG08498, RRID:CVCL_1Y51; AG07095, RRID:CVCL_0N66; AG11732, RRID:CVCL_2E35; AG04060, RRID:CVCL_2A45; AG04148, RRID:CVCL_2A55; AG04349, RRID:CVCL_2A62; AG04379, RRID:CVCL_2A72; AG04056, RRID:CVCL_2A43; AG04356, RRID:CVCL_2A69; AG04057, RRID:CVCL_2A44; AG04055, RRID:CVCL_2A42; AG13349, RRID:CVCL_2G05; AG13129, RRID:CVCL_2F55; AG12788, RRID:CVCL_L632; AG07725, RRID:CVCL_2C46; AG04064, RRID:CVCL_L624; AG04059, RRID:CVCL_L623; AG09602, RRID:CVCL_L607; AG16409, RRID:CVCL_V978; AG06234, RRID:CVCL_2B66; AG04062, RRID:CVCL_2A47; AG08433, RRID:CVCL_L625; AG16409, RRID:CVCL_V978; GM00302, RRID:CVCL_7277; AG01518, RRID:CVCL_F696; AG06234, RRID:CVCL_2B66. We routinely check all our cell cultures and confirm it to be free of mycoplasma contamination. Authentication was completed by LINE and PCR-based techniques. The International Cell Line Authentication Committee (ICLAC) lists none of these primary cells are commonly misidentified cell lines. Fibroblast cell lines were cultured and expanded in DMEM media (high glucose, Invitrogen) supplemented with 10% or 15% fetal bovine serum (Gibco), sodium pyruvate, non-essential amino acids, GlutaMAX

(Invitrogen), Pen/Strep solution, and Beta-mercaptoethanol. Fibroblast cell lines were expanded to a population doubling level (PDL) of ~19–21. Formula to calculate PDL = 3.32*log (cells harvested/cells seeded) + previous PDL. Cells were kept frozen at −150°C in the above culture medium with additional 40% FBS and 10% DMSO.

## MicroRNA-mediated neuronal reprogramming

Human fibroblasts ranging in age from 3 days to 96-year old were transduced with a lentiviral preparation of the Doxycline-inducible synthetic cluster of miR-9/9* and miR-124 (miR-9/9*-124), alongside transcription factors CTIP2, DLX1, DLX2, and MYT1L as previously described (*Richner et al., 2015*; *Victor et al., 2014*). Briefly, transduced fibroblasts were maintained in fibroblast media for two days with doxycycline prior to selection with Puromycin (3 µg/ml) and Blasticidin (5 µg/ml) at day three, then were plated onto poly-ornithine, fibronectin and laminin-coated coverslips at day five. Cells were subsequently maintained in Neuronal Media (ScienCell, Carlsbad, CA) supplemented with valproic acid (1 mM), dibutyryl cAMP (200 µM), BDNF (10 ng/ml), NT-3 (10 ng/ml), and RA (1 µM) for 30–35 days before analysis.

## Immunocytochemistry

Reprogrammed neurons were fixed with 4% paraformaldehyde (Electron Microscopy Sciences, Hatfield, PA) for 20 min at room temperature (RT), then permeabilized with 0.2% Triton X-100 for 10 min at room temperature. Cells were blocked with 1% goat serum, incubated with primary antibodies at 4°C overnight, then incubated with secondary antibodies for 1 hr at RT. Primary antibodies used for immunocytochemistry included chicken anti-MAP2 (Abcam Cat# ab5392 RRID: AB_21381531; 1:10,000 dilution), mouse anti-β-III tubulin (Covance Research Products Inc Cat# MMS-435P RRID:AB_2313773; 1:5000), rabbit anti-β-III tubulin (Covance Research Products Inc Cat# PRB-435P-100 RRID:AB_291637; 1:2000), chicken anti-NeuN (Aves Labs Cat# NUN RRID:AB_2313556; 1:500), rabbit anti-GABA (Sigma-Aldrich Cat# A2052 RRID:AB_477652; 1:2000), mouse anti-GABA (Sigma-Aldrich Cat# A0310 RRID:AB_476667, 1:500), and rabbit anti-DARPP32 (Santa Cruz Biotechnology Cat# sc-11365 RRID:AB_639000; 1:400). The secondary antibodies included goat anti-rabbit, mouse, or chicken IgG conjugated with Alexa-488, −594, or −647 (Thermo Fisher Scientific, Waltham, MA). Images were captured using a Leica SP5X white light laser confocal system with Leica Application Suite (LAS) Advanced Fluorescence 2.7.3.9723.

## Electrophysiology

Whole-cell patch-clamp recordings were performed at four weeks after transduction with miR-9/9*-124-CDM. Intrinsic neuronal properties were studied using the following solutions (in mM): Extracellular: 140 NaCl, 3 KCl, 10 Glucose, 10 HEPES, 2 CaCl$_2$ and 1 MgCl$_2$ (pH adjusted to 7.25 with NaOH). Intracellular: 130 K-Gluconate, 4 NaCl, 2 MgCl$_2$, 1 EGTA, 10 HEPES, 2 Na$_2$-ATP, 0.3 Na$_3$-GTP, 5 Creatine phosphate (pH adjusted to 7.5 with KOH). Membrane potentials were typically kept at −60 mV to −70 mV. In voltage-clamp mode, currents were recorded with voltage steps ranging from +10 mV to +80 mV. In current-clamp mode, action potentials were elicited by injection of step currents that modulated resting membrane potential from −20 mV to +80 mV. Local application of TTX (Sigma-Aldrich#T8024) was achieved using a multibarrel perfusion system with a port placed within 0.5 mm of the patched cell.

## DNA extraction

Reprogrammed neurons were harvested after 30 days of ectopic expression of miR-9/9*-124-CDM. DNA was extracted using phenol/chloroform/isoamyl alcohol followed by ethanol precipitation with a final concentration of 0.75M NaOAc and 2 µg of glycogen. DNA concentration was quantified using a standard curve with the Quant-iT dsDNA Assay Kit, broad range (Thermo Fisher Scientific, Waltham, MA) according to manufacturer's instruction, while the DNA quality was determined by the ratio of absorbance of 260 nm and 280 nm at approximately 1.7–2.0.

## Illumina DNA methylation array

The bisulfite conversion was performed for fibroblasts and reprogrammed neurons using the Zymo Research EZ-96 DNA Methylation-Gold Kit (catalog #D5008). DNA methylation data were generated

on the HumanMethylation450k Bead Chip (Illumina, San Diego, CA) according to the manufacturer's protocols. Scanning was performed via Illumina's iScan system in conjunction with the Illumina Autoloader 2 robotic arm. DNA methylation levels (β values) were established by calculating the ratio of intensities between methylated (signal A) and un-methylated (signal B) sites. The β value was calculated from the intensity of the methylated (M corresponding to signal A) and un-methylated (U corresponding to signal B) sites, as the ratio of fluorescent signals $\beta = Max(M,0)/[Max(M,0)+Max(U,0)+100]$. β values range from 0 (completely un-methylated) to 1 (completely methylated). The data were normalized using the 'Noob' normalization method (*Triche et al., 2013*).

## Epigenetic clock analysis

The epigenetic clock method is an accurate measurement of chronological age in human tissues (*Horvath, 2013*). Epigenetic age was estimated using the published software tools (*Horvath, 2013*). An online age calculator can be found at the webpage, https://dnamage. genetics.ucla.edu. The epigenetic clock has been shown to capture aspects of biological age: the epigenetic age is predictive of all-cause mortality even after adjusting for a variety of known risk factors (*Marioni et al., 2015*; *Christiansen et al., 2016*; *Horvath et al., 2015a*). The utility of the epigenetic clock method has been demonstrated in applications surrounding cognitive function (*Levine et al., 2015*), obesity (*Horvath et al., 2014*), Down syndrome (*Horvath et al., 2015b*), HIV infection (*Horvath and Levine, 2015*), and Parkinson's disease (*Horvath and Ritz, 2015*).

## Transcriptome and microRNA microarray

Total RNA was extracted from reprogrammed neurons from young fibroblasts aged three days, five months, and one year and from old fibroblast aged 90, 92, and 92 years alongside corresponding starting fibroblast samples using TRIzol (Thermo Fisher Scientific, Waltham, MA) according to the manufacturer's instruction and extracted using chloroform and ethanol precipitation. RNA quality was determined by the ratio of absorbance at 260 nm and 280 nm to be approximately 2.0. Samples for RNA microarray were then standardly prepped and labeled with Illumina TotalPrep kits (Thermo Fisher Scientific, Waltham, MA) for Agilent Human 4x44Kv1, while samples for microRNA microarray were prepared using Genisphere Flashtag labeling kits designed for Affymetrix miRNA 4.0 microarray. Standard hybridization and imagine scanning procedure were performed according to the manufacturer's protocol at Genome Technology Access Center at Washington University School of Medicine, St. Louis. The intensity of probes was imported into R environment and normalized by using package 'oligo'. Differentially expressed mRNA transcripts were identified by using package 'limma' with cut-off at adjusted p-value<0.01 and over logfc >1 fold expression change. For miRNA, the intensity of human-specific probes was isolated by using in-house python script, and were imported into R environment. Quantile normalization was performed by using 'preprocessCore' package, and differentially expressed miRNAs were identified by using package 'limma' with cut-off at adjusted p-value<0.01 and over one-fold expression change.

## Quantitative PCR validation

cDNA was generated from 4 ng of RNA using specific primer probes from TaqMan MicroRNA Assays (Thermo Fisher Scientific, Waltham, MA) and subsequently analyzed on a StepOnePlus Real-Time PCR System (AB Applied Biosystems, Foster City, CA). Expression data were normalized to RNU44 and analyzed using the $2^{-\Delta\Delta CT}$ relative quantification method. QPCR validation of miRNA expression was conducted in reprogrammed neurons from old fibroblasts aged 89, 90, 91, 92, 92, 94 compared to reprogrammed neurons from young fibroblasts aged three days, five months, one, two, 12 years of age. QPCR experiments were conducted with human striatum and human cortex slices acquired from young individuals aged 9, 11, and 19 years compared to those from older individuals aged 83, 85, and 87 years.

## MitoSOX

MitoSOX Red Mitochondrial superoxide indicator (Thermo Fisher Scientific, Waltham, MA) was diluted to 15 μM and incubated with cells for 15 min at 37°C. Cells were washed three times with PBS, dissociated with 0.25% Trypsin, then stained with DAPI. If FACs was not conducted on

the same day, cells were fixed with 4% paraformaldehyde for 20 min at room temperature. Samples were compared to untreated (unstained) fibroblast and reprogrammed neurons. Cell sorting was performed on a FACSCalibur and LsrFortessa (BD Biosciences), while quantification of the percent of the population of MitoSOX positive cells was performed using FlowJo X 10.0.7r2. Each plot on the graph represents an individual experiment with multiple reprogrammed neurons. Unpaired t-test analysis of 3 sets of experiments of reprogrammed neurons from young fibroblasts compared to reprogrammed neurons from old fibroblasts. Young samples included reprogrammed neurons from fibroblasts aged three days, five month, one year and two years, while old samples were from donors aged 86, 90, 91, 92A, and 92B years. Analyzed fibroblast samples include 1, 2, 91, 72, 74, and 94-year-old samples. P-values were calculated with the student *t*-test.

## Comet assay

Cells were prepared and analyzed using the CometAssay Kit (Trevigen) according to manufacturer's instruction. Cells were harvested after 30 days of neuronal reprogramming using 0.25% Trypsin, then whole cells were embedded in molten LMAgarose onto slides prior to overnight incubation in lysis buffer. Slides were then run in gel electrophoresis at 20 volts for 30 min, then stained with SYBR Green and visualized by epifluorescence microscopy. Tails lengths were measured by drawing a region of interest and p-values were calculated with student t-test for reprogrammed neurons from old fibroblasts aged 91, 92A and 92B compared to reprogrammed neurons from young fibroblasts aged three days, five months, and one year old, while analyzed fibroblasts were aged five months, one year, 12, 72, 86, and 92 years of age.

## Telomere analysis

Genomic DNA was collected from reprogrammed neurons from fibroblasts from one year, 56, and 86 year old donors and the corresponding fibroblasts. The isolated genomic DNA was then digested with RsaI and Hinfl and fractionated as described previously (*Tomlinson et al., 2008*). Membranes were prepared by Southern transfer and hybridized to a radioactively end-labelled (TTAGGG) 4 oligonucleotide probe as described previously (*Batista et al., 2011*).

## Acknowledgement

We thank the Genome Technology Access Center in the Department of Genetics at Washington University School of Medicine for help with processing transcriptome and microRNA microarrays. We also thank UCLA Neuroscience Genomics Core and Helen Ibsen for processing Illumina Methylation450K chips. We are grateful to Shin-ichiro Imai for helpful suggestions on the manuscript. LB is supported by the NIH K99/R00 award (4R00HL114732-03), Washington University DDRCC (NIDDK P30 DK052574) and grants from the V Foundation For Cancer Research, Edward Mallinckrodt Jr. Foundation, and the AA&MDS International Foundation. SH is supported by National Institutes of Health NIH/NIA 1U34AG051425-01 and 5R01 AG042511-02. ASY is supported by the NIH Director's Innovator Award (DP2NS083372-01) and Presidential Early Career Award for Scientists and Engineers, and grants from the Ellison Medical Foundation and Cure Alzheimer's Fund.

## Additional information

### Funding

| Funder | Grant reference number | Author |
|---|---|---|
| National Institute on Drug Abuse | R25 DA027995 | Bo Zhang |
| National Institutes of Health | K99/R00 | Luis FZ Batista |
| Washington University in St. Louis | DDRCC | Luis FZ Batista |
| National Institutes of Health | 4R00HL114732-03 | Luis FZ Batista |
| Washington University in St. Louis | NIDDK P30 DK052574 | Luis FZ Batista |

| National Institutes of Health | 1U34AG051425-01 | Steve Horvath |
| National Institutes of Health | 5R01, AG042511-02 | Steve Horvath |
| National Institutes of Health | DP2NS083372-01 | Andrew S Yoo |
| Ellison Medical Foundation | AG-NS-0878-12 | Andrew S Yoo |
| Cure Alzheimer's Fund | | Andrew S Yoo |

The funders had no role in study design, data collection and interpretation, or the decision to submit the work for publication.

### Author contributions

CJH, ASY, Conception and design, Acquisition of data, Analysis and interpretation of data, Drafting or revising the article, Contributed unpublished essential data or reagents; BZ, Conception and design, Analysis and interpretation of data, Drafting or revising the article; MBV, LFZB, SH, Acquisition of data, Analysis and interpretation of data, Drafting or revising the article; SD, Analysis and interpretation of data, Drafting or revising the article, Contributed unpublished essential data or reagents

### Author ORCIDs

Andrew S Yoo, http://orcid.org/0000-0002-0304-3247

## Additional files

### Major datasets

The following dataset was generated:

| Author(s) | Year | Dataset title | Dataset URL | Database, license, and accessibility information |
| --- | --- | --- | --- | --- |
| Christine J Huh, Steve Horvath, Bo Zhang, Andrew S Yoo | 2016 | Datasets from: Matintenance of age in human neurons generated by microRNA-based neuronal conversion of fibroblasts, DNA methylation and annotation, transcriptome and microRNAs | http://dx.doi.org/10.5061/dryad.t6096 | Available at Dryad Digital Repository under a CC0 Public Domain Dedication |

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
