## [Decision Letter]

Thank you for submitting your manuscript "Maintenance of age in human neurons generated by micro-RNA-based neuronal conversion of fibroblasts" to *eLife*. Three experts reviewed your manuscript, and their assessments, together with my own, Jeremy Nathans (the Reviewing editor), form the basis of this letter. As you will see, overall, the reviewers were impressed with the importance of your work.

I am including the three reviews at the end of this letter, as there are a variety of specific and useful comments in them. One part of the data that appears problematic is the comet assay. See the detailed comments by reviewer #1. A second point is the extent to which your work is distinct from or is an advance beyond that reported by Mertens et al. [Cell Stem Cell 17: 705-718. (2015)]. See the comments by reviewer #2. A fuller description of the Mertens et al. data and its relation to yours would be useful for the reader. This could be done in the Discussion section.

*Reviewer #1:*

Huh et al. used microRNA based reprogramming to generate neurons from fibroblasts, starting with fibroblasts taken from humans over a range of ages, and claim that the reprogrammed neurons reflect the age of the donor based on epigenetic assays, DNA damage assays, gene expression patterns, microRNA expression patterns, telomere length, and other parameters. They convincingly show that the reprogramming works with cells of a range of ages, showing good staining of neuron-specific markers, and impressive changes in gene expression patterns from a fibroblast pattern to a neuronal pattern. They use a variety of assays to support the interpretation that the conversion maintains the "age" of the donor cells. In general, the data support their conclusions well. They use a DNA methylation assay as an assay of "epigenetic age" and show a reasonable correlation between fibroblast age and epigenetic age as measured in the converted neurons.

In their gene expression analysis, the PCA analysis seems to nicely separate converted neurons from fibroblasts as expected, but does not clearly separate older from younger cells of either type. The analysis of microRNA expression patterns does seem to separate older from younger of each type, though based upon a very small number of microRNA.

Regarding their GO analysis (Figure 3) of differentially expressed genes, they appear to refer to "GO terms associated with aging"; some of these terms obviously relate to aging (for example, aging), while many others are not obviously related to aging (negative regulation of metabolic processes, nervous system development, etc.) and it is not clear to me or well explained in the text where these come from or how they are related to aging per se. Please explain.

Comet assay: I am not an expert, but I found this unconvincing. There appears to be a big difference in loading that at least contributes to, if not accounts for, the claimed difference in comet tail length between young and old neurons. Both of the young assays (I assume these are "young" re-programmed neurons) seem underloaded compared to the older cells; and if we were to control in our mind's eye for this loading difference, I am not sure the lengths of the comet tails would really differ much or at all. At best, the difference seems extremely subtle, much less than is illustrated in the figure.

This figure would be stronger if the young re-programmed neurons were run against the young fibroblasts, and then old fibroblasts run against old neurons-this might be what is illustrated but the figure legend does not give enough information to understand it. But then they need to load equal amounts of DNA in each experiment, or explain why the assay is such that this is not a problem.

*Reviewer #2:*

This manuscript is a follow-up to a previous work from the same authors in which neuronal micro RNAs (miRNAs) were used to directly reprogram human fibroblasts into specific neuronal subtypes. The goal of this paper is to demonstrate that miRNA-derived neurons retain the age profile of their donors. Interestingly, fibroblasts from donors sampling a wide age range (from a few days old to 94 years old), after being reprogrammed into neurons retained the DNA methylation levels, telomere size, transcription profile, levels of DNA damage, etc. This result is in contrast to neurons differentiated from induced pluripotent stem cells (iPSCs) which reset their clock in terms of telomere size, gene expression profile, oxidative stress, etc.

This manuscript is straightforward and technically sound. The authors were thorough in testing all the important age markers and convincingly demonstrate the key point of the paper. However, this work is purely a cell characterization study and therefore only incremental (the actual miRNA conversion itself has been published in several articles before). More importantly, it has already been shown that fibroblasts reprogrammed into neurons (via a different route) retain the age-specific transcriptomic profile, see Mertens et al., Directly Reprogrammed Human Neurons Retain Aging-Associated Transcriptomic Signatures and Reveal Age-Related Nucleocytoplasmic Defects, Cell Stem Cell, 2015. This published paper has only casually been referenced. Along the same lines, in the Introduction, the study was motivated by the following sentence: "Because this neuronal conversion is direct and bypasses pluripotent / multipotent stem cell stages, we reasoned that directly reprogrammed neurons would retain the age signature of the original donor." This statement makes it seem as if this phenomenon has not already been shown before.

At this stage the paper is therefore only incremental and does not warrant publication in *eLife*. The miRNA-converted neurons could indeed potentially be a powerful tool in studying age-related diseases (just like the reprogramming method published by Mertens et al) and so, at this stage, it would be very valuable to already see a functional comparison between examples of iPSC and miRNA-generated neurons from a diseased patient or a more thorough study on the onset and specific causes of age-related changes.

A comment regarding the data in Figure 4: the data points for the 1-yo and 56-yo examples are not clear and the retention of telomere lengths of fibroblasts and miRNA-generated neurons does not appear convincing. That said, the 86-yo point, which indeed is the most important point, is convincing. Moreover, a more detailed description of the image is warranted in the legend, for instance, what do the numbers on the left stand for?

*Reviewer #3:*

This manuscript builds on prior work from the Yoo lab to test whether miRNA coverted neuronal cells derived from fibroblasts retain age-related signatures. I have a few comments regarding this work:

1) It would be of interest to quantitatively examine the expression levels of INK4A locus genes (p16 and ARF) since their upregulation has also been associated with aging.

2) Regarding telomere rejuvenation upon iPS reprogramming, since the referenced papers were published (Agarwal et al., 2010 and Marion et al., 2009), it has been shown that the original telomere length is only incrementally reset with iPS reprogramming (Batista Nature 2011, Moon Nat Gen 2015 and several others). This should be corrected.

3) It may be important to acknowledge that the retention of molecular marks of aging may make it more difficult to do in vitro with these cells. For example, shorter telomere length from older individuals may limit the replicative potential. Along the same lines, I suppose it's important to acknowledge that there may be other features of aging that were not assayed here that may be altered during the conversion process.

---

## [Author Response]

*I am including the three reviews at the end of this letter, as there are a variety of specific and useful comments in them. One part of the data that appears problematic is the comet assay. See the detailed comments by reviewer #1. A second point is the extent to which your work is distinct from or is an advance beyond that reported by Mertens et al. [Cell Stem Cell 17: 705-718. (2015)]. See the comments by reviewer #2. A fuller description of the Mertens et al. data and its relation to yours would be useful for the reader. This could be done in the Discussion section.*

Thank you for pointing out important comments to be addressed in the revision. As you suggested, we addressed reviewer #1’s comments about the comet assay by conducting additional experiments in young and old fibroblasts, and providing further details about technical details employed in the assay. We also provided an extra paragraph illustrating the advances our paper makes beyond the paper by Mertens et al. in the Discussion as suggested by reviewer #2. We furthermore conducted additional analysis to address an interesting question posed in comment #1 by reviewer #3.

*Reviewer #1:*

*Huh et al. used microRNA based reprogramming to generate neurons from fibroblasts, starting with fibroblasts taken from humans over a range of ages, and claim that the reprogrammed neurons reflect the age of the donor based on epigenetic assays, DNA damage assays, gene expression patterns, microRNA expression patterns, telomere length, and other parameters. They convincingly show that the reprogramming works with cells of a range of ages, showing good staining of neuron-specific markers, and impressive changes in gene expression patterns from a fibroblast pattern to a neuronal pattern. They use a variety of assays to support the interpretation that the conversion maintains the "age" of the donor cells. In general, the data support their conclusions well. They use a DNA methylation assay as an assay of "epigenetic age" and show a reasonable correlation between fibroblast age and epigenetic age as measured in the converted neurons.*

*In their gene expression analysis, the PCA analysis seems to nicely separate converted neurons from fibroblasts as expected, but does not clearly separate older from younger cells of either type. The analysis of microRNA expression patterns does seem to separate older from younger of each type, though based upon a very small number of microRNA.*

We agree that the separation between older and younger cells of either cell type is not clearly demonstrated on this 2D PCA plot. We therefore replaced the previous 2D PCA plots with a 3D PCA plot that more clearly illustrates the segregation of samples not only based on cell type (fibroblast vs. reprogrammed neurons), but also based on age (young vs. old). We hope this PCA plot provides more clear evidence to show the transcriptomic differences between young and old fibroblasts and between young and old reprogrammed neurons.

Thank you for pointing out the number of microRNAs the PCA is based on. The bar graph in Figure 3 represents only the commonly upregulated and downregulated microRNAs with age observed in both fibroblasts and reprogrammed neurons; however, the PCA not only includes these commonly regulated microRNAs, but also includes microRNAs that change with age uniquely in reprogrammed neurons and in fibroblasts. We highlight the commonly changed microRNAs in Figure 3 to illustrate that the age-regulated microRNAs seen in fibroblasts is conserved after direct conversion in reprogrammed neurons.

*Regarding their GO analysis (Figure 3) of differentially expressed genes, they appear to refer to "GO terms associated with aging"; some of these terms obviously relate to aging (for example, aging), while many others are not obviously related to aging (negative regulation of metabolic processes, nervous system development, etc.) and it is not clear to me or well explained in the text where these come from or how they are related to aging per se. Please explain.*

We agree that we did not clearly explain the age-association of the GO terms for Figure 3—figure supplement 1. We have now included references that relate the cellular processes of select GO terms represented in the bar graph to aging in the Results and Discussion sections of the text.

*Comet assay: I am not an expert, but I found this unconvincing. There appears to be a big difference in loading that at least contributes to, if not accounts for, the claimed difference in comet tail length between young and old neurons. Both of the young assays (I assume these are "young" re-programmed neurons) seem underloaded compared to the older cells; and if we were to control in our mind's eye for this loading difference, I am not sure the lengths of the comet tails would really differ much or at all. At best, the difference seems extremely subtle, much less than is illustrated in the figure.*

*This figure would be stronger if the young re-programmed neurons were run against the young fibroblasts, and then old fibroblasts run against old neurons-this might be what is illustrated but the figure legend does not give enough information to understand it. But then they need to load equal amounts of DNA in each experiment, or explain why the assay is such that this is not a problem.*

Thank you for your suggestion. We agree that the differences with age in fibroblasts are important data to illustrate. We have now included the fibroblast data for the comet assay in Figure 4. Likewise, we also supplemented Figure 4 with fibroblast data for the MitoSOX experiment as well. We updated the “Results and Discussion” section of the text and figure legend to reflect these changes. To address the concern regarding the loading amounts, we agree we did not clearly explain in the text the concept of the comet assay. We have now updated our text to describe the comet assay as a single cell analysis. To this end, we load 1x10^5^ whole live cells onto agar then place onto a coverslip. After the cells are embedded in low melting-point agar, we incubate the coverslip in lysis buffer overnight, then we run it through gel electrophoresis to allow a voltage to run through. Each comet represents the DNA content of one cell and the more DNA damage that exists the further the DNA will travel, creating longer comets. Since we aren’t loading DNA specifically and rather whole individual cells, it is unlikely that the differences in comet tail length are attributable to technique such as loading error. We additionally stained the cells with SYBR green and visualized the cells with the same exposure, gain, and fluorescent intensity. We have also updated the figure legend to provide more information to better explain what we represent in the figure.

*Reviewer #2:*

*This manuscript is a follow-up to a previous work from the same authors in which neuronal micro RNAs (miRNAs) were used to directly reprogram human fibroblasts into specific neuronal subtypes. The goal of this paper is to demonstrate that miRNA-derived neurons retain the age profile of their donors. Interestingly, fibroblasts from donors sampling a wide age range (from a few days old to 94 years old), after being reprogrammed into neurons retained the DNA methylation levels, telomere size, transcription profile, levels of DNA damage, etc. This result is in contrast to neurons differentiated from induced pluripotent stem cells (iPSCs) which reset their clock in terms of telomere size, gene expression profile, oxidative stress, etc.*

*This manuscript is straightforward and technically sound. The authors were thorough in testing all the important age markers and convincingly demonstrate the key point of the paper. However, this work is purely a cell characterization study and therefore only incremental (the actual miRNA conversion itself has been published in several articles before). More importantly, it has already been shown that fibroblasts reprogrammed into neurons (via a different route) retain the age-specific transcriptomic profile, see Mertens et al., Directly Reprogrammed Human Neurons Retain Aging-Associated Transcriptomic Signatures and Reveal Age-Related Nucleocytoplasmic Defects, Cell Stem Cell, 2015. This published paper has only casually been referenced. Along the same lines, in the Introduction, the study was motivated by the following sentence: "Because this neuronal conversion is direct and bypasses pluripotent / multipotent stem cell stages, we reasoned that directly reprogrammed neurons would retain the age signature of the original donor." This statement makes it seem as if this phenomenon has not already been shown before.*

*At this stage the paper is therefore only incremental and does not warrant publication in eLife. The miRNA-converted neurons could indeed potentially be a powerful tool in studying age-related diseases (just like the reprogramming method published by Mertens et al) and so, at this stage, it would be very valuable to already see a functional comparison between examples of iPSC and miRNA-generated neurons from a diseased patient or a more thorough study on the onset and specific causes of age-related changes.*

We have now included a more thorough discussion of the aforementioned paper in our Discussion. Specifically, we have added sentences about the microRNA-based conversion approach (which is a different method than the one used in the Mertens et al. study) reaching similar conclusions regarding the maintenance of age-associated transcriptomic changes of starting fibroblasts in reprogrammed neurons. We ensured that we provided proper references throughout the manuscript when we discuss transcriptome changes. However, transcriptome analysis is only a fraction of what we assessed in our study, and we respectively argue that our findings about epigenetic clock measurements and microRNA profiles (as well as cellular properties) are novel and important findings. Aging is a complex biological process and with the implication of utilizing directly converted human neurons to model late-onset diseases, it is important to characterize many hallmarks of aging to achieve a more comprehensive sense of the age status of the reprogrammed neurons, which has been accomplished in our study. The future goals in our lab are to investigate the contribution and role of age-regulated microRNAs in aging.

*A comment regarding the data in Figure 4: the data points for the 1-yo and 56-yo examples are not clear and the retention of telomere lengths of fibroblasts and miRNA-generated neurons does not appear convincing. That said, the 86-yo point, which indeed is the most important point, is convincing. Moreover, a more detailed description of the image is warranted in the legend, for instance, what do the numbers on the left stand for?*

Thank you for your comments. We have updated our figure to indicate the numbers on the left of our image reflect the kilobase length. We also included the following explanation in our text to “Additionally, neuronal conversion maintained the telomere length of starting fibroblasts, which is virtually unchanged, (Figure 4) in contrast to the progressive increase in length commonly observed with iPSC reprogramming, where telomeres reach a plateau of around 12-14 kilobases after a few cellular passages (Agarwal et al., 2010; Batista et al., 2011; Marion et al., 2009).”

*Reviewer #3:*

*This manuscript builds on prior work from the Yoo lab to test whether miRNA coverted neuronal cells derived from fibroblasts retain age-related signatures. I have a few comments regarding this work:*

*1) It would be of interest to quantitatively examine the expression levels of INK4A locus genes (p16 and ARF) since their upregulation has also been associated with aging.*

It is interesting to pursue whether CDKN2A levels change in fibroblasts or in reprogrammed neurons with age. To answer this question, we illustrate the age-associated fold-change of CDKN2A in fibroblasts and in reprogrammed neurons (according to our transcriptome profiling data in Figure 2). We further conducted a qPCR using two different primer sets to validate the microarray with qPCR.

Author response image 1.**DOI:**
http://dx.doi.org/10.7554/eLife.18648.014

Additionally, we also conducted immunostaining analysis of CDKN2A (Figure 6). Collectively, we did not detect significant alteration in CDKN2A expression with aging by microarray and qPCR assays in both fibroblasts and converted neurons. However, we do detect a marked increase in CDKN2A in reprogrammed neurons compared to starting fibroblasts, perhaps a reflection of cells adopting a post-mitotic fate.

Author response image 2.**DOI:**
http://dx.doi.org/10.7554/eLife.18648.015

*2) Regarding telomere rejuvenation upon iPS reprogramming, since the referenced papers were published (Agarwal et al., 2010 and Marion et al., 2009), it has been shown that the original telomere length is only incrementally reset with iPS reprogramming (Batista Nature 2011, Moon Nat Gen 2015 and several others). This should be corrected.*

Thank you for your comments. We have updated our text to “Additionally, neuronal conversion maintained the telomere length of starting fibroblasts, which is virtually unchanged, (Figure 4) in contrast to the progressive increase in length commonly observed with iPSC reprogramming, where telomeres reach a plateau of around 12-14 kilobases after a few cellular passages (Agarwal et al., 2010; Batista et al., 2011; Marion et al., 2009).”

*3) It may be important to acknowledge that the retention of molecular marks of aging may make it more difficult to do in vitro with these cells. For example, shorter telomere length from older individuals may limit the replicative potential. Along the same lines, I suppose it's important to acknowledge that there may be other features of aging that were not assayed here that may be altered during the conversion process.*

Thank you for this suggestion. We have incorporated these potential limitations in the Conclusion section of the revised manuscript.